# Mechanistic Pathways and Molecular Targets of Plant-Derived Anticancer *ent*-Kaurane Diterpenes

**DOI:** 10.3390/biom10010144

**Published:** 2020-01-16

**Authors:** Md. Shahid Sarwar, Yi-Xuan Xia, Zheng-Ming Liang, Siu Wai Tsang, Hong-Jie Zhang

**Affiliations:** 1School of Chinese Medicine, Hong Kong Baptist University, Kowloon Tong, Hong Kong, China or xiayixuan@hkbu.edu.hk (Y.-X.X.); chem@outlook.com (Z.-M.L.); tsang@hkbu.edu.hk (S.W.T.); 2Department of Pharmacy, Noakhali Science and Technology University, Sonapur, Noakhali 3814, Bangladesh

**Keywords:** *isodon* genus, *ent*-kaurane diterpenoids, cancer, natural compounds, pathways

## Abstract

Since the first discovery in 1961, more than 1300 *ent*-kaurane diterpenoids have been isolated and identified from different plant sources, mainly the genus *Isodon*. Chemically, they consist of a perhydrophenanthrene subunit and a cyclopentane ring. A large number of reports describe the anticancer potential and mechanism of action of *ent*-kaurane compounds in a series of cancer cell lines. Oridonin is one of the prime anticancer *ent*-kaurane diterpenoids that is currently in a phase-I clinical trial in China. In this review, we have extensively summarized the anticancer activities of *ent*-kaurane diterpenoids according to their plant sources, mechanistic pathways, and biological targets. Literature analysis found that anticancer effect of *ent*-kauranes are mainly mediated through regulation of apoptosis, cell cycle arrest, autophagy, and metastasis. Induction of apoptosis is associated with modulation of BCL-2, BAX, PARP, cytochrome c, and cleaved caspase-3, -8, and -9, while cell cycle arrest is controlled by cyclin D1, c-Myc, p21, p53, and CDK-2 and -4. The most common metastatic target proteins of *ent*-kauranes are MMP-2, MMP-9, VEGF, and VEGFR whereas LC-II and mTOR are key regulators to induce autophagy.

## 1. Introduction

Cancer is a heterogeneous disease which caused 9.6 million deaths globally in 2018 and the incidence of cancer is projected to increase to 13.1 million deaths by 2030 [1,2]. In addition, a large number of cancer patients have forms of the disease that are resistant to the mainstay anticancer drugs. Currently used anticancer drugs are mainly obtained from natural sources (plants, microorganisms, and marine organisms) or from synthetic reactions [3]. Unfortunately, most of these drugs also kill normal cells generating severe side effects [4,5]. These factors indicate the urgent need for the development of new therapeutics and treatment strategies.

Medicinal plants are an important source of drug entities. Over the past few decades, plants became an indispensable source of anticancer agents which are commonly used to treat different types of cancers in clinical settings. The World Health Organization (WHO) estimates that 80% of the total world’s population residing in rural areas rely on plant sources as a part of their primary health care system. In addition, more than one-quarter of the prescribed drugs in developed countries are prepared directly or indirectly from plants [6]. In developing countries, 75% of the population use plant-derived drugs for the treatment of various illnesses [7]. Since the early 1900s, thousands of medicinal plants have been thoroughly investigated to screen cytotoxic compounds, yielding numerous anticancer drugs and lead compounds [5]. The structure of the cytotoxic plant metabolite podophyllotoxin was first described in 1932 [8], this was followed by the discovery of vinca alkaloids (vinblastine and vincristine) in 1958 [9], and later, the identification of paclitaxel in 1971 [10]. These plant-derived molecules have revolutionized cancer treatment, yet more lead structures continue to be needed.

*ent*-Kaurane diterpenoids have recently attracted the attention of scientists and doctors due to their promising anticancer effect and safety profile [11]. They are subclass of tetracyclic diterpenes which consist of a perhydrophenantrene unit (A, B, and C rings) and cyclopentane unit (D ring) connected by a bridge of two carbons between C-8 and C-13 (Figure 1). Diterpenes containing a kaurane scaffold with configurational inversion at all chiral centers are known as *ent*-kaurane diterpenes. In this case, the prefix “*ent*-” is added before the name of this class of compounds to indicate that they are enantiomers of kaurene diterpenes. Moreover, International Union of Pure and Applied Chemistry (IUPAC) has established structured-guidelines for the stereochemistry, nomenclature, and numbering of the *ent*-kaurane skeleton. With the exception of those compounds containing an alkene functionality between C-9 and C-11, most of *ent*-kauranes show characteristic negative values for the specific optical rotation [12]. This class of compounds are mainly available in *Isodon* genus which mainly consists of a group of flowering plants. Other plant species that also contain *ent*-kaurane compounds belong to different families such as Asteraceae, Lamiaceae, Euphorbiaceae, Jungermanniaceae, and Chrysobalanaceae [13]. Accumulating evidences described the anticancer activity and underlying mechanism of *ent*-kaurane compounds against numerous cancers such as lung, colon, breast, prostate, liver, and gastric cancer both in vitro and in vivo. [14,15]. In this article, we discuss the anticancer *ent*-kaurane diterpenoids reported in the literature along with their source and the biological targets regulated by this type of compound.

### Methodology

Literature search was performed through the commonly used databases such as PubMed, MEDLINE, Scopus, Embase, Google Scholar, ScienceDirect, and Wiley Online. The articles which described the anticancer activity, mechanism of action and protein targets of *ent*-kaurane diterpenes were included in the review whilst those papers which mentioned only the isolation and characterization without further exploration of their molecular pathways were excluded. The molecular structures of diterpenoids were generated using the ChemDraw software. The chemical formulae, molecular weight, and related information of the diterpenoids were comprehensively matched with SciFinder^®^, a well-known portal for chemical compounds.

## 2. Anticancer *ent*-Kauranes and Their Targets

### 2.1. Oridonin

Oridonin (C_20_H_28_O_6_, MW: 364.44, Figure 2), was first isolated from *Isodon rubescens* in the 1960s and one of its derivatives l-alanine-(14-oridonin) ester trifluoroacetate (HAO472) (Figure 2) has progressed to phase-I clinical trials by Hengrui Medicine Co. Ltd., Lianyungang, China [16]. Numerous studies reported the anticancer activities of oridonin against diverse human cell lines such as HCT116 (colorectal carcinoma, IC_50_: 32.6 μM) [17], OCI-AML3 (acute myeloid leukemia, IC_50_: 3.27 μM) [18], and BxPC-3 (pancreatic carcinoma, IC_50_: 53.0 µM) [19] after 24 h treatment; KYSE-150 (esophageal carcinoma, IC_50_: 28.7 μM), EC9706 (esophageal carcinoma, IC_50_: 34.4 μM), KYS-30 cells (esophageal carcinoma, IC_50_: 32.3 μM) [20], Jurkat (T-cell leukemia, IC_50_: 0.73  μM) [21], MG-63 (osteosarcoma, IC_50_: 10.9 μM), HOS (osteosarcoma, IC_50_: 11.9 μM), Saos-2 (osteosarcoma, IC_50_: 17.3 μM), U-2OS cells (osteosarcoma, IC_50_: 17.7 μM) [22], EC109 (esophageal carcinoma, IC_50_: 19.7 μM), EC9706 (esophageal carcinoma, IC_50_: 31.3 μM), EC1 (esophageal carcinoma, IC_50_: 25.8 μM) [23], HUVECs (human umbilical vein endothelial cells, IC_50_: 420 µM) [24], CNE1 (nasopharyngeal carcinoma, IC_50_: 3.66 μM), and CNE2 (nasopharyngeal carcinoma, IC_50_: 5.93 μM) cells [25] after 48 h treatment; and EC109 (esophageal carcinoma, IC_50_: 67.1 μM), SHG-44 (glioblastoma, IC_50_: 53.5 μM), MCF-7 (breast carcinomas, IC_50_: 72.1 μM) [26], and SGC-7901 (gastric carcinoma, IC_50_: 22.7 µM) [27] after 72 h treatment. Kadioglu et al. (2018) tested the cytotoxicity of oridonin in spectrum of drug-resistant cancer cell lines including CCRF-CEM (leukemia, IC_50_: 1.65 μM), CEM/ADR5000 (leukemia, IC_50_: 8.53 μM), MDA-MB231 (breast carcinoma, IC_50_: 6.06 μM), MDA-MB231/BCRP (breast carcinoma, IC_50_: 9.74 μM), HCT116 (p53^+/+^) (IC_50_: 18.0 μM), HCT-116 (p53^−/−^) (IC_50_: 34.7 μM), U87MG (glioblastoma, IC_50_: 17.4 μM), HepG2 (liver carcinoma, IC_50_: 25.7 μM), and AML12 (liver normal cells, IC_50_: >109 μM) cells, but did not mention the incubation time with oridonin in these cytotoxic test experiments [28].

A wide variety of sources have reported the anticancer mechanism of oridonin in multiple cancer cell lines [17,24,27,29,30,31,32]. According to the study of Yao et al. (2017) oridonin induced autophagy by decreasing the protein levels of glucose transporter 1 (GLUT1) and monocarboxylate transporter 1 (MCT1) in SW480 human colorectal cancer cells, in the BALB/c xenograft model. Autophagy induction by oridonin was correlated to increased expression of light chain-I (LC-I) and LC-II, while decreasing the phosphorylation of adenosine monophosphate-activated protein kinase (p-AMPK) [17]. In human umbilical vein endothelial cells (HUVECs), oridonin suppressed the proliferation, tube formation, migration, and invasion, but induced apoptosis. Oridonin was also found to block the angiogenesis of zebrafish by decreasing the mRNA expression of vascular endothelial growth factor A (VEGFA), vascular endothelial growth factor receptor 2 (VEGFR2) and VEGFR3, while reducing the expression of metastatic proteins claudin-1, -4, and -7. Antimetastatic activity of oridonin was further confirmed in a xenograft zebrafish model tested at a dose of 8 mg/kg that showed significant effect (*p* < 0.01) in comparison to the vehicle control group [24]. Gao et al. (2016) reported that oridonin induced apoptosis by downmodulating the expression level of B-cell lymphoma 2 (BCL-2), but upmodulating the expression of BAX, thus reducing the BCL-2/BAX ratio in human gastric cancer SGC-7901 cells. Moreover, oridonin treatment activated caspase-3 by promoting the release of cytochrome c from mitochondria to the cytosol [27]. In another study, oridonin potentiated the anticancer activity of lentinan, a polysaccharide isolated from shiitake mushrooms, by upregulating the expression levels of caspase-3, -8, -9, BAX, p53, and p21 while downregulating the expression of BCL-2, B-cell lymphoma extra-large (BCL-X_L_) and epidermal growth factor (EGF) in SMMC-7721 human hepatoma cells [29].

In prostate cancer cells (PC-3 and DU-145), oridonin increased the expression levels of p53, p21, caspase-3, -9, and poly (ADP-ribose) polymerase (PARP), while it decreased cyclin-dependent kinase 1 (CDK1) levels [30]. Moreover, it inhibited the expression of phosphoinositide 3-kinase (PI3K) and blocked phosphorylation of protein kinase B (p-Akt). Sun et al. (2018) reported synergistic anticancer activity of oridonin in combination with lentinan by decreasing the expression of BCL-2 and nuclear factor kappa B (NF-κB) and increasing the expression of caspase-3, -9, p53, p21, NF-κB inhibitor-α (IκB-α) by increasing transcription of mRNA and translation of the mRNA into their respective protein products in human hepatoblastoma HepG2 cells [31]. In 4T1 human breast cancer cells, oridonin inhibited cellular proliferation, migration, and invasion via a negative modulation of notch1-4. Furthermore, the administration of oridonin (5 mg/kg) significantly (*p* < 0.01) reduced the weight (84%) and volume (72%) of 4T1 tumors compared with the negative control group in a xenograft nude mouse study [32].

### 2.2. Eriocalyxin B

Eriocalyxin B (C_20_H_24_O_5_, MW: 344.41, Figure 2) was isolated from the plant *Isodon eriocalyx* var. laxiflora [33]. The cytotoxicity of eriocalyxin B was tested in panel of human cell lines such as SMMC-7721 (hepatocarcinoma, IC_50_: 0.76 μM) [33], MCF-7 (IC_50_: 0.75 μM), MDA-MB-231 (IC_50_: 0.47 μM) [34] after 48 h treatment; PANC1 (pancreatic carcinoma, IC_50_: 1.79 μM), CAPAN1 (pancreatic carcinoma, IC_50_: 0.86 μM), CAPAN2 (pancreatic carcinoma, IC_50_: 0.73 μM), SW1990 (pancreatic carcinoma, IC_50_: 1.40 μM), WRL68 (normal human liver cells, IC_50_: >3.58 μM), PBMC (human peripheral blood mononuclear cells, IC_50_: >5.83 μM) [35], SU-DHL-4 (lymphoma, IC_50_: 1.00 μM), Namalwa (lymphoma, IC_50_: 1.50 μM), Raji (lymphoma, IC_50_: 2.00 μM), Jurkat (lymphoma, IC_50_: 2.00 μM), U266 (lymphoma, IC_50_: 5.60 μM) and HUT78 (lymphoma, IC_50_: 2.50 μM) cells [36] after 72 h treatment.

The molecular pathways of eriocalyxin B against different cancers have been intensively investigated [33,34,37,38,39]. In hepatocellular carcinoma SMMC-7721 cells, eriocalyxin B induced apoptosis by interfering with the binding of NF-κB with the response elements via targeting cysteine 62 moiety of p50 [33]. In a recent study, eriocalyxin B was reported to induce autophagy upon an upregulation of LC3B-II and beclin-1 in MCF-7 and MDA-MB-231 human breast cancer cells; however, the expression of p62 was downregulated. In addition to autophagy, eriocalyxin B also induced apoptosis by activating caspase-3 and PARP while decreasing BCL-2. In fact, the phosphorylation levels of Akt, mammalian target of rapamycin (mTOR), and p70S6K were decreased in a dose- and time-dependent manner post eriocalyxin B treatment. Therefore, the mechanism of action of eriocalyxin B is suggested to be associated with the Akt/mTOR/p70S6K signaling pathway [34].

Earlier studies demonstrated that eriocalyxin B induced G1 phase cell cycle arrest by decreasing the expression of cyclin D1, CDK4, and phosphorylated retinoblastoma (p-Rb). When applied to HUVECs at 50 and 100 nM, eriocalyxin B notably inhibited the VEGF-induced cell proliferation, tube formation, cell migration, and invasion [37]. Further to its modulation on the VEGF cascade, eriocalyxin B suppressed the VEGF-induced phosphorylation of VEGFR-2 via an interaction with various ATP-binding sites, thus leading to the repression of several VEGFR-2 downstream molecules such as VEGFR-1, focal adhesion kinase (FAK), Src, pSer^473^-Akt, extracellular signal-regulated kinase (ERK1/2), and pThr^180^/Tyr^182^-mitogen-activated protein kinase (p38-MAPK). When administered in vivo, eriocalyxin B restrained the formation of new blood vessels, vascularization, and growth of the 4T1 breast tumor xenografts. According to the study of Lu et al. (2016), eriocalyxin B inhibited the cell proliferation, migration, invasion, and angiogenesis of human colon cancer cells SW1116 [38]. Mechanistic studies found that it inhibited the phosphorylation of janus kinase 2 (JAK2) and signal transducer and activator of transcription-3 (STAT3) as well decreased the expression of VEGF, VEGFR-2, matrix metallopeptidase-2 (MMP-2), MMP-9, and proliferating cell nuclear antigen (PCNA). Yu et al., (2015) reported that eriocalyxin B inhibited both constitutive- and interleukin-6 (IL-6)-induced phosphorylation of STAT-3 in A549 lung cancer cells without affecting the upstream kinases such as JAK1, JAK2, and tyrosine kinase 2 (TYK2) [39].

### 2.3. Excisanin A

Excisanin A (C_20_H_30_O_5_, MW: 350.455, Figure 2) was isolated from *Isodon macrocalyxin* which exhibited cytotoxic potential in breast cancer MDA-MB-231 (IC_50_: 22.4 μM), SKBR3 (IC_50_: 27.3 μM) [40] and MDA-MB-453 (IC_50_: 10.3 μM) cells as well as in hepatoma Hep3B (IC_50_: 6.45 μM) cells [41] after 72 h treatment. At 10–40 μM concentration, excisanin A suppressed the expression of MMP-2, MMP-9, integrin β1, β-catenin, and reduced the phosphorylation of FAK and Src in breast cancer (MDA-MB-231 and SKBR3) cells. Moreover, the phosphorylation of PI3K, Akt and glycogen synthase kinase 3β (GSK3β) was also reduced after treatment with excisanin A [40]. In addition to its effect on breast cancer metastasis, excisanin A inhibited the proliferation of human hepatocellular carcinoma cell line Hep3B and breast cancer cell line MDA-MB-453 cells by inducing apoptosis. It significantly (*p* < 0.01) reduced the tumor weight (46.4%) as well as induced tumor cell apoptosis in Hep3B xenograft mice at a dose of 20 mg/kg/day. Mechanistic studies showed that excisanin A reduced the expression of p-GSK-3α/β, Thr308-Akt, Ser473-Akt, p-mTOR, and p-FKHR expression in both Hep3B and MDA-MB-453 cells [41]. Zhang et al. (2013) reported dose- and time-dependent induction of autophagy by increasing the expression of LC3-II and decreasing the expression of p62 in nasopharyngeal carcinoma (NPC) cell lines CNE1 and CNE2 cells after excisanin A treatment. Moreover, excisanin A also upregulated p-JNK (c-jun N-terminal kinase), p-c-Jun and sestrin 2 expression in NPC cells [42].

### 2.4. Ponicidin

Ponicidin (C_20_H_26_O_6_, MW: 362.42, Figure 2) was isolated from *Isodon rubescens* and *I. japonicas*, and showed cytotoxic activity against the human carcinoma cell lines K562 (leukemia), Bcap37 (breast), GC823 (gastric), BIU87 (bladder), HeLa (cervical) and PC-3 (prostate) cells [43]. Further studies also described the cytotoxicity of ponicidin in HeLa (IC_50_: 23.1 μM) [44], A549 (IC_50_: 38.0 μM) and GLC-82 (lung, IC_50_: 32.0 μM) cancer cell lines after 24 h; A549 cells (IC_50_: 31.0 μM) and GLC-82 (IC_50_: 26.0 μM) after 48 h; A549 cells (IC_50_: 15.0 μM) and GLC-82 cells (IC_50_: 13.0 μM) [45] after 72 h. It suppressed the growth of gastric carcinoma MKN28 cells in dose- and time-dependent manner by downregulating BCL-2, p-JAK2, and p-STAT3 expression, while upregulating BAX and cleaved caspase-3 expression [43]. An earlier study reported induction of apoptosis and disruption of mitochondrial membrane potential in ponicidin-treated lung cancer A549 and GLC-82 cells by the upregulating expression of cleaved caspase-3, -8, -9, and BAX, and downregulating the expression of BCL-2 and survivin [45]. Du et al., (2015) reported that ponicidin induced apoptosis and cell cycle arrest in colon cancer cells HCT-116 by increasing caspase-3, BAX, and p-p38 expression, while decreasing BCL-2, p-ERK, and p-Akt expression [46].

### 2.5. Pharicins A and B

Pharicins A (C_24_H_34_O_7_, MW: 434.53, Figure 2) and B (C_24_H_34_O_8_, MW: 450.53) were isolated from *Isodon pharicus*. The only structural difference between these two compounds is the position of hydroxyl group which is located at C-1 in pharicin A and at C-12 in pharicin B. Pharicin A induced mitotic arrest in leukemia cell line Jurkat and solid tumor-derived cell line raji by inhibiting auto phosphorylation activity of BubR1 [47]. Investigation of the effects of pharicin B in several acute myeloid leukemia (AML) cell lines and primary leukemia cells from AML patients such as NB-4, U937 and THP-1 cells showed cytotoxicity with IC_50_ values around 3.5 μM after 48 h of treatment. Further studies found that it can quickly stabilize the retinoic acid receptor-α (RAR-α) protein, even in the presence of all-trans-retinoic acid (ATRA) which generally induce the loss of RAR-α protein. Moreover, pharicin B also increased ATRA-dependent transcriptional activity of RAR-α protein in NB4 cells, a type of promyelocytic leukemia–RAR-α–positive APL cell line [48].

### 2.6. Jaridonin

Jaridonin (C_22_H_32_O_5_, MW: 376.49, Figure 2) was isolated from the Chinese herb *Isodon rubescens* which exhibited cytotoxicity in esophageal squamous cancer EC109 (IC_50_: 12.0 μM), EC9706 (IC_50_: 11.2 μM), and EC1 (IC_50_: 4.60 μM) cells [23] after 48 h as well as in human glioma cell line SHG-44 (IC_50_: 14.7 μM) and breast cancer MCF-7 (IC_50_: 16.7 μM) cell [26] after 72 h. A mechanistic study found that it induced apoptosis and G2/M phase cell cycle arrest in esophageal cancer EC109, EC9706 and EC1 cells. Jaridonin treatment caused remarkable reduction of mitochondrial membrane potential, release of cytochrome c into the cytosol, and increased expression of caspase-3 and -9, leading to activation of the mitochondria-mediated apoptosis. Moreover, jaridonin also increased the production of reactive oxygen species (ROS) and upregulated p53, p21waf1/Cip1, and BAX expression [23]. Another study confirmed the induction of G2/M phase arrest in esophageal squamous cancer EC9706 cells [49]. However, cell cycle arrest was related with increased phosphorylation of ataxia-telangiectasia mutated (ATM) (Ser1981) protein kinase, cell division control 2 (Cdc2) (Tyr15), Cdc25C, and H2A histone family member X (H2A.X)(Ser139) as well as increased expression of check point kinase 1(Chk1) and Chk2.

### 2.7. Jungermannenones A and B

Jungermannenones A (C_20_H_28_O_2_, MW: 300.44, Figure 2) and B (C_20_H_28_O_3_, MW: 316.44, Figure 2) were isolated from *Jungermannia fauriana*, and exhibited cancer cell killing activity in multiple cell lines [50]. The IC_50_ values of jungermannenone A were determined in PC3 (IC_50_: 1.34 μM), DU145 (prostate carcinoma, IC_50_: 5.01 μM), LNCaP (prostate carcinoma, IC_50_: 2.78 μM), A549 (IC_50_: 8.64 μM), MCF-7 (IC_50_: 18.3 μM), HepG2 (IC_50_: 5.29 μM) and RWPE1 (normal prostate epithelial, IC_50_: 5.09 μM) cell lines, while the IC_50_ values of jungermannenone B were measured in PC3 (4.93 μM), DU145 (5.50 μM), LNCaP (3.18 μM), A549 (5.26 μM), MCF-7 (14.2 μM), HepG2 (6.02 μM), and RWPE1 (18.2 μM). Both compounds induced apoptosis by stimulating ROS accumulation and inducing cell cycle arrest. Western blot analysis showed reduced expression of c-myc, cyclin D1, cyclin E, CDK4, and p-Cdc2, while increased expression of p21 and p-ERK. Furthermore, both jungermannenones A and B induced DNA damage by reducing the expression of DNA repair proteins Ku70/Ku80 and RDA51 [50].

### 2.8. Effusanin E

Effusanin E (C_20_H_28_O_6_, MW: 364.44, Figure 2) was isolated from *Rabdosia serra* (*Rabdosia* is a synonym of *Isodon*), and is cytotoxic to nasopharyngeal carcinoma cell CNE2 (IC_50_: ~60 μM) but non-toxic up to 500 μM in normal nasopharyngeal epithelial cell line after 48 h. It inhibited cell proliferation and induced apoptosis in CNE2 cells by increasing the expression of cleaved PARP, caspase-3 and -9 proteins, as well as decreasing nuclear translocation of p65 NF-κB [51]. Moreover, the binding ability of NF-κB to the promoter region of cyclooxygenase-2 (COX-2) abolished after treating the cells with effusanin E, therefore, inhibiting the expression and promoter activity of COX-2. In vivo studies also found significant reduction of tumor growth (*p* < 0.05) at 30 mg/kg without any sign of toxicity in comparison to negative control group (DMSO) as well as decreased expression of p50 NF-κB and COX-2 in tumor tissue.

### 2.9. Longikaurin A

Longikaurin A (C_20_H_28_O_5_, MW: 348.44, Figure 2) was described from *Isodon ternifolius*, and displayed cytotoxic potential in human hepatocarcinoma SMMC-7721 (IC_50_: ~1.8 μM), HepG2 (IC_50_: ~2 μM), BEL7402 (IC_50_: ~6 μM), Huh7 (IC_50_: ~6 μM), and LO2 (IC_50_: ~9 μM) cells [52], and the nasopharyngeal carcinoma CNE1 (IC_50_: 1.26 μM) and CNE2 (IC_50_: 1.52 μM) cells [25] with 48 h incubation. This *ent*-kaurane compound induced apoptosis and G2/M phase cell cycle arrest in SMMC-7721 and HepG2 cells. In vivo studies found longikaurin A treatment (6 mg/kg) significantly suppressed tumor development (*p* < 0.01) in a SMMC-7721 xenograft mouse model study and the antitumor effect was comparable to the positive control 5-FU (fluorouracil) (10 mg/kg). A mechanistic study showed reduced expression of S-phase kinase-associated protein 2 (Skp2) after longikaurin A treatment, which was correlated with increased expression of p21 and p-Cdc2 (Try15) and decreased expression of cyclin B1 and Cdc2 proteins [52]. It also induced ROS production and phosphorylation of JNK.

In nasopharyngeal carcinoma (NPC) cell lines S18 and S26, longikaurin A exerted anticancer activity and suppressed the stemness of cells. Western blot analysis showed downregulation of c-myc and fibronectin in NPC cells [53]. At a low concentration (0.2 μM), longikaurin A induced S phase arrest while at a higher concentration (3.12 μM) it induced apoptosis in human NPC cell lines CNE1 and CNE2 cells [25]. Mechanistic studies found upregulation of cleaved caspase-3, cleaved PARP, and BAX, while downregulation of BCL-xL, p-Akt, and p-GSK-3β. In CNE2 xenograft model, longikaurin A significantly suppressed tumor growth without affecting the body weights of the mice [53].

### 2.10. Glaucocalyxins A and B

Glaucocalyxin A (C_20_H_28_O_4_, MW: 332.44, Figure 2) was isolated from *Isodon japonica*, and demonstrated strong anticancer effects in the human cell lines of HL-60 (leukemia, IC_50_: 6.15 µM) cells [54] after 24 h; Focus (hepatocarcinoma, IC_50_: 2.70 μM), SMMC-7721 (IC_50_: 5.58 μM), HepG2 (IC_50_: 8.22 μM), SK-HEP1 (hepatocarcinoma, IC_50_: 2.87 μM) [55], HOS (IC_50_: 7.02 μM), Saos-2 (IC_50_: 7.32 μM), U-2OS (IC_50_: 8.36 μM), MG-63 (IC_50_: 5.30 μM) [56], and UMUC3 (bladder carcinoma, IC_50_: 9.77 μM) cells [57] after 48 h; and MCF-7 (IC_50_: 1.00 μM) and Hs578T (breast carcinoma, IC_50_: 4.00 μM) [58] after 72 h.

Glaucocalyxin A dose- and concentration-dependently inhibited the growth of the liver cancer Focus and SMMC7721 cells. Mechanistic studies found induction of G2/M phase cell cycle arrest as well as increased expression of cleaved caspase-3 and PARP [55]. In MCF-7 and Hs578T breast cancer cells, glaucocalyxin A induced apoptosis, and G2/M phase arrest by increasing the expression of intrinsic apoptotic markers cleaved caspase-3, BAX, and p53, while decreasing BCL-2 expression. It also upregulated the expression of extrinsic apoptotic markers such as Fas and Fas ligand (FasL) at the mRNA and protein level. Moreover, the expression of p-ERK and p-JNK increased in a dose-dependent manner after glaucocalyxin A treatment [58]. Furthermore, it concentration-dependently suppressed the proliferation and induced apoptosis in human brain glioblastoma U87MG cells by activating caspase-3, while downregulating p-Akt, p-Bad, and X-linked inhibitor of apoptosis protein (XIAP) [59]. In another study, glaucocalyxin A induced a dose-dependent apoptosis in HL-60 cells by increasing the expression of caspase-3, -9, and BAX, while decreasing the expression of BCL-2. It caused loss of mitochondrial membrane potential and release of cytochrome c from mitochondria into the cytosol as well as elevated intracellular ROS generation [60].

Glaucocalyxin B (C_22_H_30_O_5_, MW: 374.48, Figure 2), which differs from glaucocalyxin A by an acetylation of the hydroxyl group at C14, exerted anticancer activity in leukemia cell line HL-60 (IC_50_: 5.86 µM) after 24 h [54], gastric cancer cell line SGC-7901 (IC_50_: 13.4 µM) [61] after 60 h, and cervical cancer cell lines HeLa (IC_50_: 4.61 μM) and SiHa (IC_50_: 3.11 μM) after 72 h [62]. The anticancer activity in HeLa and SiHa cells was related with induction of apoptosis and autophagy by increasing the expression of phosphatase and tensin homolog (PTEN) protein and cleaved PARP, as well as increasing the cleavage of LC3 II/I protein. Moreover, glaucocalyxin B also reduced the expression of p-Akt in HeLa and SiHa cells [62].

### 2.11. Lasiodin

Lasiodin (C_22_H_30_O_7_, MW: 406.48, Figure 2) was isolated from *Isodon serra*, and inhibited the proliferation and migration of human nasopharyngeal carcinoma cells CNE1 (IC_50_: ~6 μM) and CNE2 (IC_50_: ~5 μM) at 24 h incubation [63]. Mechanistic studies showed that lasiodin-induced apoptosis was related to increased expression of apoptotic protease activating factor 1 (Apaf-1), release of cytochrome c, and cleavages of PARP, caspase-3 and -9. Moreover, the expression of p-Akt, p-ERK1/2, p-p38, and p-JNK was also reduced after 24 h treatment with lasiodin. However, lasiodin-mediated inhibition of cell proliferation was blocked in the cells treated with Akt or MAPK inhibitors. Lasiodin treatment also downregulated the expression of COX-2, revoked NF-κB binding to the COX-2 promoter and stimulated the nuclear translocation of NF-κB. Therefore, lasiodin inhibited the proliferation of CNE1 and CNE2 cells by activating Apaf-1/caspase-dependent apoptotic pathways and suppressing Akt/MAPK and COX-2/NF-κB signaling cascades.

### 2.12. Adenanthin

Adenanthin (C_26_H_34_O_9_, MW: 490.55, Figure 2) was isolated from the leaves of *Isodon adenanthus*, and showed antiproliferative effects against hepatocellular carcinoma cells such as HepG2 (IC_50_: 2.31 μM), Bel-7402 (IC_50_: 6.67 μM), and SMMC-7721 (IC_50_: 8.13 μM) while exhibiting less toxicity in the two normal hepatic cell lines QSG-7701 (IC_50_: 19.6 μM) and HL-7702 cells (IC_50_: 20.4 μM) at 48 h exposure [64]. At 72 h treatment, the IC_50_ values of adenanthin in esophageal carcinoma cell line EC109, glioma cell line SHG-44, and breast cancer cell line MCF-7 were 6.50, 4.80, and 7.60 μM, respectively [26]. According to the study of Hou et al. (2014), adenanthin killed malignant liver cells by stimulating ROS generation and targeting peroxiredoxin (Prx) I and II proteins which are considered essential for survival of HCC cells. In-vivo studies with SMMC-7721 cells xenograft mice showed significantly reduced tumor size at 10 mg/kg without any notable side effects. However, the body weight of the mice decreased after treatment with 20 mg/kg of adenanthin [64]. In a different study, the same research group also found that adenanthin stimulated acute promyelocytic leukemia (APL) cell differentiation by targeting Prx I and Prx II as well as blocking their peroxidase activities. Moreover, the level H_2_O_2_ increased after adenanthin treatment, which led to activation of p-ERK, p-c-Jun, and increased transcription of CCAAT-enhancer-binding protein β (C/EBPβ) [65].

### 2.13. Kaurenic Acid

Kaurenic acid or kaurenoic acid (KA) (C_21_H_32_O_2_, MW: 316.49, Figure 2), chemically known as *ent*-kaur-16-en-19-oic acid, was isolated from *Espeletia semiglobulata*, and exhibited antimelanoma effects with an IC_50_ value of 0.79 μM in B16F1 cells. An in vivo study found that KA (160 mg/kg) markedly (*p* < 0.001) reduced the tumor sizes (49.51%) compared with the vehicle control group (87.5%) in a C57BL/6 mice model. RT-PCR analysis showed reduced expression of BCL-xL at the mRNA level after KA treatment in C57BL/6 mice [66]. In another study, KA selectively reduced the cell viability of two breast cancer cells MCF7 (proficient P53) and SKBR3 (mutated p53), while KA treatment was resistant to HB4A cell line [67]. At 70 μM, KA caused 40% and 25% cell death of MCF7 and SKBR3 cells suggesting that p53 protein may play a vital role in anticancer activity of KA. A recent study reported significant genotoxicity, apoptosis, and cell cycle arrest in gastric cancer cells after KA treatment [68]. At 10 μg/mL concentration, KA downregulated the expression of c-myc, CCND1, BCL-2, and caspase-3, while it upregulated ATM, Chk2, and TP53 expression.

### 2.14. Weisiensin B

Weisiensin B (C_20_H_28_O_5_, MW: 348.44, Figure 2) was isolated from the traditional Chinese herb *Isodon weisiensis* and inhibited the growth and proliferation of human hepatoma cell lines BEL-7402 and HepG2, ovarian cancer HO-8910 cells, and gastric cancer SGC-7901 cells [69]. The IC_50_ values of the compound were 10.0, 3.24, 32, and 4.34 μM in BEL-7402, HepG2, HO-8910 cells and SGC-7901 cells, respectively, after 48 h. DNA fragmentation assay and Hoechst 33,258 staining showed that weisiensin B significantly induced apoptosis. Flow cytometry analysis with propidium iodide (PI) staining revealed induction of G2/M phase arrest after weisiensin B treatment. Another study reported weisinensis B-mediated induction of apoptosis, G2/M phase arrest, and significant ROS generation in human chronic myeloid leukemia K562 cells [70].

### 2.15. Inflexinol

Inflexinol (C_24_H_34_O_8_, MW: 450.53, Figure 2) was isolated from *Isodon excisus*, which is native to China, Korea, and Japan. It inhibited the growth of the colon cancer cell lines SW620 (IC_50_: 29.0 μM), HCT116 (p53^+/+^) (IC_50_: 30.0 μM) and HCT116 (p53^−/−^) (IC_50_: 34.0 μM) but did not show toxicity in the normal colon CCD-112 CoN cells up to 40 μM [71]. Antiproliferative effects of inflexinol were mediated through induction of apoptosis via downregulation of antiapoptotic markers BCL-2, XIAP, and cIAP1/2, with simultaneous upregulation of cleaved caspase-3, -9, and PARP. Inflexinol treatment also significantly increased the ratio of BAX/BCL-2 and suppressed the expression of cyclin D1 and BCL-2. Moreover, inhibitory effects of inflexinol on the growth of colon cancer cells were related to inactivation of NF-κB by modification of a cysteine residue in the p50 subunit. An in vivo study in a SW620 xenograft model showed dose-dependent reduction of tumor volumes (64.5% to 47.8%) and weights (75.9% to 58%) at 12 and 36 mg/kg, respectively, compared with the control group. Moreover, it also exhibited inhibition of DNA binding activity of NF-κB in tumor tissue and suppressed nuclear translocation of p65 and p50 as well as inhibited the phosphorylation of IκB in the cytosol [71].

### 2.16. Xerophilusin B

Xerophilusin B (C_20_H_26_O_5_, MW: 346.42, Figure 2) was isolated from *Isodon xerophilus* and showed dose-dependent growth inhibition of esophageal squamous cell carcinoma KYSE-140 (IC_50_: 2.80 μM), KYSE-150 (IC_50_: 1.20 μM), KYSE-450 (IC_50_: 1.70 μM), and KYSE-510 (IC_50_: 2.60 μM) cells when incubated for 72 h. Further studies found that xerophilusin B induced apoptosis and G2/M phase cell cycle arrest in KYSE-150 and KYSE-450 cells [72]. Treatment with xerophilusin B increased the release of mitochondrial cytochrome c and upregulated the expression of cleaved caspase-3 and -9, while downregulated caspase-7 and PARP levels. Moreover, the ratio of BCL-2/BAX decreased after xerophilusin B treatment. An in vivo study in BALB/c nude mice showed significant inhibition of tumor growth at 15 mg/kg without any major adverse effects.

### 2.17. Henryin

Henryin (C_22_H_32_O_6_, MW: 392.49, Figure 2) was isolated from *Isodon rubescens*, and inhibited the proliferation of colorectal cancer SW480 (IC_50_: 0.27 μM), HT-29 (IC_50_: 0.77 μM), and HCT-116 cells (IC_50_: 0.90 μM), and lung cancer A549 cells (IC_50_: 2.47 μM), but is slightly less toxic to normal colon cells CCD-841-CoN (IC_50_: 2.98 μM) and normal bronchus cells BEAS-2B (IC_50_: 3.55 μM). The inhibitory effects of henryin were correlated to reduced expression of cyclin D1 and c-myc. Moreover, it inhibited the binding of β-catenin to transcription factor 4 (TCF-4) [73].

### 2.18. 11α, 12α-epoxyleukamenin E

11α, 12α-epoxyleukamenin E (EPLE) (C_22_H_30_O_6_, MW: 390.48, Figure 2) was isolated from *Salvia cavaleriei*, and exhibited cytotoxic activity in colorectal cancer cell lines HCT-116 and SW480 by downregulating the markers of the Wnt-signaling pathway such as c-myc, axin2 and survivin as well as inhibiting β-catenin transcriptional activity [74]. It induced apoptosis by decreasing the expression of BCL-2 and Bcl-xL, while increasing the expression of Bim and caspase-3. Moreover, a combination of EPLE and 5-fluorouracil produced synergistic anticancer effects in colon cancer cells. An in vivo study in the xenograft model also showed significant reduction of tumor size after EPLE treatment [74].

### 2.19. DEK

*Ent*-kaur-2-one-16β,17-dihydroxy-acetone-ketal (DEK) (C_23_H_36_O_3_, MW: 360.54, Figure 2) was isolated from the leaves of *Rubus corchorifolius*, and exhibited cytotoxic activity (IC_50_ = 40.0 μM) in HCT-116 human colon cancer cells after 72 h treatment but was non-toxic up to 100 μM in human colonic myofibroblasts CCD-18Co cells [75]. Mechanistic studies found that DEK induced apoptosis by increasing the expression of cleaved caspase-3, -9, PARP, p53, BAX, and p21Cip1/Waf1, while decreasing the expression of cell cycle markers such as cyclin D1, CDK2, and CDK4. Moreover, the expression of two carcinogenic proteins including epidermal growth factor receptor (EGFR) and COX-2 were reduced and the activation of Akt was inhibited after DEK treatment.

### 2.20. JDA-202

JDA-202 (C_21_H_30_O_5_, MW: 362.47, Figure 2) was isolated from *Isodon rubescens*, and exhibited anticancer effects in esophageal cancer EC109 (IC_50_: 8.60 μM) and EC9706 (IC_50_: 9.40 μM) cells, yet was considerably non-toxic in normal cell lines KYSE-450 (IC_50_: 26.2  μM) and HET-1A (IC_50_: 36.1 μM) at 24 h drug incubation. Antiproliferative activity was related to direct binding of JDA-202 to the antioxidant protein Prx I, inhibition of its activity and expression, as well as induction of hydrogen peroxide (H_2_O_2_)-related cell death in esophageal cancer cell lines. Moreover, JDA-202 treatment also led to the increased phosphorylation of JNK, p38, and ERK, which were suppressed by ROS scavenger *N*-acetylcysteine (NAC) and H_2_O_2_ scavenger catalase. Intravenous administration of JDA-202 at 20 mg/kg/day in Balb/c nude mice significantly inhibited the tumor growth without loss of body weight and organ toxicities [26].

### 2.21. Pterisolic Acid G

Pterisolic acid G (PAG) (C_20_H_28_O_6_, MW: 364.44, Figure 2), isolated from *Pteris semipinnata*, dose- and time-dependently inhibited the growth of human colorectal cancer cell line HCT-116 with IC_50_ values 20.4, 16.2, and 4.07 µM for 24, 48, and 72 h, respectively. Mechanistic studies found that PAG reduced the expression of disheveled segment polarity protein 2 (Dvl-2), GSK-3β, β-catenin, cyclin D1, and c-myc in HCT-116 cells. Moreover, it induced apoptosis by increasing the expression of p53, puma, cleaved PARP, and cleaved caspase-3, as well as decreasing the expression of p-p65, BCL-2, and BCL-xL [76].

### 2.22. Rabdoternin B and Maoecrystal I

Rabdoternin B (C_21_H_30_O_7_, MW: 394.46, Figure 2) and maoecrystal I (C_22_H_30_O_8_, MW: 422.47, Figure 2) were isolated from the aerial parts of *Isodon rosthornii* [77] and the leaves of *I. xerophilus* [78], respectively. After 48 h, rabdoternin B showed cytotoxicity in colon cancer SW480 (IC_50_: 23.2 µM), HT-29 (IC_50_: 36.3 µM), and HCT-116 (IC_50_: 20.7 µM) cells, but no toxicity in normal colon CCD-841-CoN (IC_50_: >40 µM) cells. The IC_50_ values of maoecrystal I were 16.2, 11.4, 26.2, and >40 µM in SW480, HT-29, HCT-116, and CCD-841-CoN cells, respectively. Both compounds induced G2/M phase arrest in SW480 cells. Further research with maoecrystal I found that its anticancer activity was related with downregulation of Wnt-signaling target genes such as c-myc, cyclin D1, surviving, and axin2 [79].

### 2.23. CHKA

Huang et al. (2016) isolated 3α-cinnamoyloxy-9β-hydroxy-*ent*-kaura-16-en-19-oic acid (CHKA) (C_29_H_36_O_5_, MW: 464.6, Figure 2) from the ethanol extract of *Wedelia chinensis*. CHKA from petroleum ether fraction showed most potent anti-angiogenic activity in zerbrafish model. Moreover, CHKA treatment also considerably blocked a series of VEGF-stimulated angiogenesis events such as proliferation, invasion, and tube formation of endothelial cells. Additionally, it also diminished the activity of VEGFR-2 tyrosine kinase and inhibited several downstream targets including p-VEGFR-2, p-mTOR, p-Akt, and p-ERK in HUVECs. In a mouse model, CHKA significantly blocked sprouts formation of the aortic ring, and inhibited subsequent formation of vessels [80].

### 2.24. CrT1

*Ent*-18-acetoxy-7β-hydroxy kaur-15-oxo-16-ene (CrT1) (C_22_H_32_O_4_, MW: 360.49, Figure 2) was isolated from the traditional Vietnamese medicinal plant *Croton tonkinensis*, and exhibited dose- and time-dependent antiproliferative activity in various cancer cell lines with IC_50_ values ranging between 8.40 and 31.2 µM but no toxicity in fibroblast NIH-3T3 cells (IC_50_ not reported) [81]. CrT1 induced apoptosis and G1 cell cycle arrest in human hepatocellular carcinoma SK-HEP1 cells by the activation of caspase-3, -7, -8, -9, and PARP. Moreover, the expression of p53 and BAX increased, while the expression of BCL-2 decreased after CrT1 treatment. CrT1 also increased the cytoplasmic translocation of cytochrome c, upregulated the expression of p-AMPK, while downregulating the expression of p-mTOR and p-p70S6K.

### 2.25. Ent-16β-17α-dihydroxykaurane

*Ent*-16β-17α-dihydroxykaurane (DHK) (C_20_H_34_O_2_, MW: 306.49, Figure 2) was isolated from the bark of *Croton malambo*, and exerted pro-apoptotic effects in human breast cancer cell line MCF-7 (IC_50_: 40.8 µM) after 72 h by downregulating BCL-2 expression at both the mRNA and protein levels, as well as decreasing human telomerase reverse transcriptase (hTERT) expression at the mRNA level [82]. The same research group later found dissociation of activator protein 2 alpha (Ap2α)–Rb activating complex in MCF-7 cells after DHK treatment, which affects the binding ability of the complex to BCL-2 gene promoter. This process leads to downregulation of BCL-2 as well as upregulation of E2F transcription factor 1 (E2F1) and its target pro-apoptotic gene puma [83].

### 2.26. Ent-11α-hydroxy-16-kauren-15-one

*Ent*-11α-hydroxy-16-kauren-15-one (KD) (C_20_H_30_O_2_, MW: 302.46, Figure 2) was isolated from the Japanese liverwort *Jungermannia truncata*, and induced apoptosis in human promyelocytic leukemia HL-60 (IC_50_: 0.56 µM) cells through activation of caspase-8. Treatment with KD also resulted in activation of caspase-9 but caspase-9-specific inhibitor could not diminish KD-induced apoptosis. Moreover, KD treatment resulted in time-dependent cleavage of caspase-8-substrate Bid as well as proteolytic processing of procaspase-8, suggesting association of caspase-8-dependent pathway in KD-induced apoptosis [84].

### 2.27. Hydroxy-15-oxo-zoapatlin

Hydroxy-15-oxo-zoapatlin (OZ) (C_20_H_26_O_4_, MW: 330.42, Figure 2) was isolated from plants in the genus *Parinari* and has been thoroughly investigated for its anticancer activity. Evaluation of this molecule for its proapoptotic activity in acute lymphoblastic leukemia Molt4 (IC_50_: 5.00 µM) cells showed that OZ induced the externalization of hypodiploidia and phosphatidylserine which are considered two hallmarks of apoptosis. Furthermore, OZ treatment also increased the expression of PARP and caspase-3 in Molt4 cells [85].

The most commonly reported anticancer pathways of the *ent*-kaurane diterpenoids with their biological targets from different pathways have been summarized in Appendix A and Figure 3.

## 3. Conclusions

Nature has blessed humanity with a wide variety of drugs for the treatment of cancer, and over the last few decades a large number of natural products have been widely investigated for their anticancer potential. In recent years, *ent*-kaurane diterpenoids have drawn great attention from the scientific community to evaluate their cancer inhibiting effects. Despite the identification of mechanistic pathways of a large number of compounds in both in vitro and in vivo models, there still remains much to understand about their mechanism of action, direct drug binding activity, and most importantly, to design and/or conduct preclinical and clinical research. We hope this article will provide a new platform for encouraging thorough investigation of *ent*-kaurane diterpenoids for the development of new anticancer drug candidates to treat the growing number of cancer patients. Although we have discussed the anticancer activity and mechanistic pathways of several *ent*-kaurane diterpenes, there remain many compounds with similar structures that we have not included that are certainly worthy of further exploration as anticancer agents. Additional work on the structure–activity relationship (SAR) of the *ent*-kaurane diterpenoids with their various molecular targets could also shed light as a way towards bringing some of these potent anticancer molecules into the clinic.

## Figures and Tables

**Figure 1 biomolecules-10-00144-f001:**
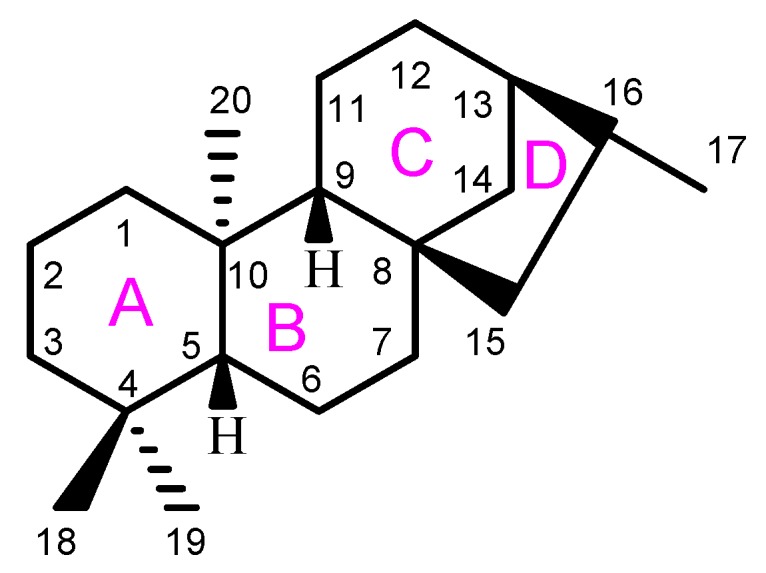
General structure and numbering system of *ent*-kaurane diterpenes.

**Figure 2 biomolecules-10-00144-f002:**
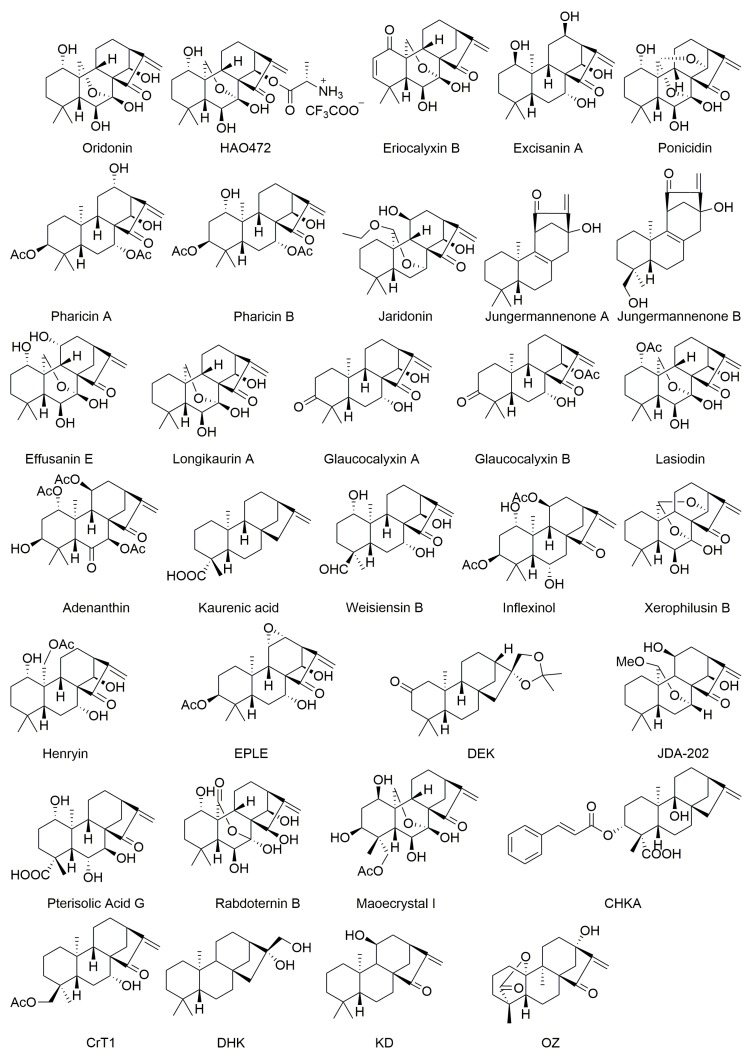
Structures of plant-derived anticancer *ent*-kaurane diterpenoids.

**Figure 3 biomolecules-10-00144-f003:**
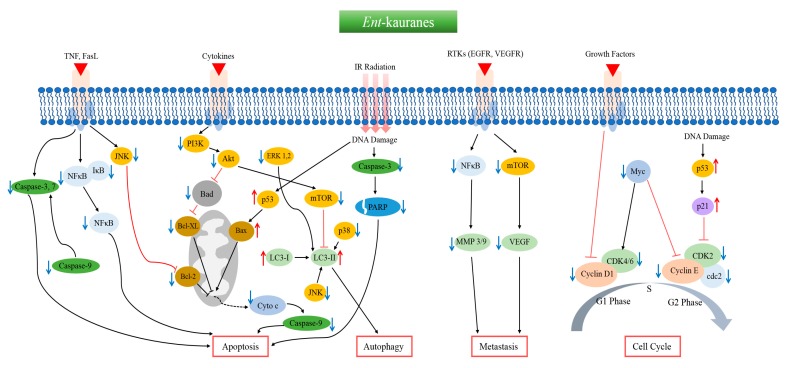
The common biological targets and mechanistic pathways of anticancer *ent*-kaurane diterpenoids.

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
