# Peer review of "Mechanistic Pathways and Molecular Targets of Plant-Derived Anticancer ent-Kaurane Diterpenes"

_biomolecules, 2020, doi:10.3390/biom10010144_

Round 1

Reviewer 1 Report

The rewiew by Zhang et al. is interesting, overall well-written, update and complete. I've only some minor suggestion: 

Please, add a short description of your literature search and selection strategy Please include a short discussion section listing criticism and perspective on the topic, in order to avoid that it seems only a systematic list of studies Since the review is not systematic, probably the authors could focus on main studies, limiting the citation from less known and local journals

Reviewer 2 Report

This manuscript describes the biological activities and mechanisms of action of 27 ent-kaurane diterpenes, isolated from plant species, in particular from Isodon genus.

The manuscript falls in the scope of Biomolecules, however, some corrections are necessary in order to improve the paper. Furthermore, English revision by a native speaker is mandatory.

Abstract should be reformulated focusing on the goal of the paper and the most important compounds described.

Line 14 – please correct:  “Chemically, most of them consist of a perhydrophenanthren subunit and a cyclopentane ring.” This is the basic skeleton of kaurane diterpenes, all of kaurane diterpenes have this scaffold.

Lines 15 – 16 -  “…reported the anticancer potential and mechanism of action of ent-kaurane compounds in treatments of different cancers.” Since there are no ent-kaurane diterpenes in clinical therapy for oncological diseases, this sentence must be reformulated.

Lines 19-26 – there is no sense in writing this type of considerations in the abstract. Please delete.

Line 27 - correct: Isodon genus (instead of Isodon plants)

Lines 31 – 32 – this sentence needs more recent and appropriate references. Which are the data from WHO?

Line 41 – statement needs reference from WHO

Lines 51 – 54 – some biogenetic considerations should be included. The differences between kaurane and ent-kaurane diterpenes should be careffuly described.

Lines 58 – 62 – in the introduction section it seems that the goal of this review will be the description of bioactive ent-kaurane diterpenes isolated from Isodon species; this should be corrected and clarified since authors only described 27 compounds and  several of them were not isolated from Isodon genus. Therefore, this part of the introduction should be reformulated.

Line 74 – compounds were isolated from plant extracts; authors should correct this all over the paper; the proper term should be “isolated” not “extracted”.

Lines 129 – since the manuscript concerns ent-kaurane diterpenes, there is no need in saying and repeating that in the beginning of the description of all the compounds; please correct all over the manuscript.

Line 220 – the chemical difference between pharacins A and B should be described.

Line 212 – the correct sentence should be “jaridonin was isolated from …”

Line 367 – “Xerophilusin B is a plant-derived bioactive compound” – since all compounds described in this paper were isolated from plants, this sentence should be re-written.

Line 432 – focus should be done on CHKA diterpene and not on the 12 ent-kauranes that were isolated. Please correct accordingly.

Quality of all chemical structures and figure 3 must be improved.
